# Quantifying Regional Contributions to Sex Classification from Fundus Photographs via a Two-Stage Attention-Based Deep Learning Approach

**Shina Jang**[1,3]                                              SAJANG@INHA.AC.KR
**Joseph Kim**[1,2,*]                                          JKIM0529@NAVER.COM

[1] *Inha University College of Medicine, Incheon, Republic of Korea*

[2] *Department of Ophthalmology, Inha University Hospital, Incheon, Republic of Korea*

[3] *Department of Obstetrics and Gynecology, Inha University Hospital, Incheon, Republic of Korea*

[*] *Corresponding author*

**Editors:** Accepted for publication at MIDL 2026

## Abstract

Existing explainability methods for fundus-based sex classification rely on qualitative saliency visualizations, making it difficult to objectively quantify the proportional contribution of anatomical regions. We propose a two-stage multi-branch framework using pre-trained ResNet50 backbones for region-specific feature extraction from three predefined retinal ROIs (macula, optic disc, and vasculature), combined with an attention-based fusion module that produces ROI-level scalar weights. In a single-center dataset of 3,478 eye-level fundus images (1,973 subjects), the fusion model achieved the highest discriminative performance (AUC = 0.861; 95% CI: 0.826–0.895), significantly outperforming all single-branch models ($p \leq 0.038$). The fusion mechanism assigned comparable weights to the macula (0.408; 95% CI: 0.369–0.446) and optic disc (0.383; 95% CI: 0.347–0.418; $p = 0.506$), both significantly higher than the vasculature (0.209; 95% CI: 0.184–0.238; $p < 0.001$). These quantitative, reproducible region-level weights offer a methodological step toward anatomically interpretable explainability in fundus-based classification.

**Keywords:** fundus photograph, sex classification, explainable AI, attention mechanism, retinal imaging

## 1. Introduction

Deep learning models predict biological sex from fundus photographs with high accuracy (Poplin et al., 2018; Korot et al., 2021), yet the anatomical basis remains opaque. Prior explainability studies using Grad-CAM (Betzler et al., 2021) and patch-based saliency (Ilanchezian et al., 2021) consistently implicate the optic disc, macula, and vasculature, but yield qualitative heatmaps that preclude reproducible, quantitative comparison of regional contributions. We address this gap with a structurally explicit multi-branch attention framework that outputs ROI-level scalar fusion weights with uncertainty estimates—enabling, for the first time, statistically rigorous quantification of region-wise contributions to sex classification from fundus photographs.

## 2. Methods

**Dataset.** Conventional fundus photographs were collected from 1,973 subjects (1,000 male, 973 female). Subject-wise partitioning yielded a training set (1,741 subjects; 3,078 images) and a held-out validation set (232 subjects; 400 images), each balanced at a 1:1 male-to-female ratio at the image level.

**Regional image extraction.** Three ROIs were isolated per image: vasculature via a U-Net pre-trained on the FIVES dataset; optic disc via brightness-based localization (crop = 25% of shorter image dimension); and macula via fovea localization relative to the disc center (crop = 2× disc ROI). All ROIs were resized to 224 × 224 pixels and normalized using ImageNet statistics.

**Model architecture.** As shown in Figure 1, three parallel ResNet50 branches (ImageNet initialization) extract L2-normalized feature vectors from each ROI. An attention MLP (two hidden layers: 256 and 128 units; batch normalization, ReLU, dropout $p = 0.3$) produces three scalar fusion weights via softmax:

$$A_i = \frac{\exp(L_i)}{\sum_{j=1}^{3} \exp(L_j)}$$

The weighted sum $Z_{\text{fused}} = \sum_i A_i \cdot Z_i$ is classified by a single linear layer. Weights $A_i$ serve as quantitative ROI-level indicators of model reliance.

**Training.** Each branch was pre-trained independently for 30 epochs (Adam, lr=$10^{-5}$; binary cross-entropy; best-AUC checkpoint retained), after which backbones were frozen and only the fusion MLP and classifier were trained for 30 epochs under identical settings. Early stopping (patience=5) and augmentation (flipping, rotation $\leq 30°$, color jittering) were applied.

**Statistical analysis.** All metrics and fusion weights were estimated with 95% CIs via paired bootstrapping (2,000 resamples); $p < 0.05$ was considered significant.

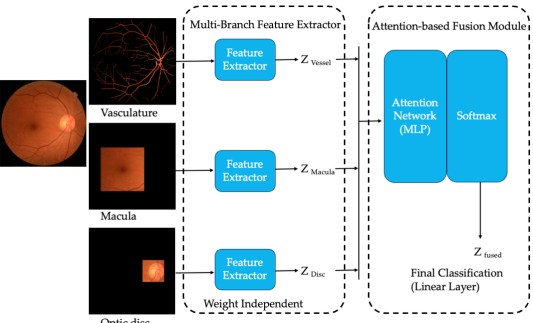

Figure 1: Overview of the proposed two-stage multi-branch fusion framework. Stage 1 independently pre-trains three ROI-specific ResNet50 branches. Stage 2 freezes the backbones and trains the attention fusion module to produce ROI-level scalar weights $A_i$ and the final sex prediction.

## 3. Results

**Classification performance.** The fusion model achieved the highest AUC (0.861; 95% CI: 0.826–0.895) and accuracy (0.772), significantly outperforming all single-branch models ($p \leq 0.038$; Table 1).

Table 1: Sex classification performance. All 95% CIs estimated via paired bootstrapping (2,000 resamples). $p$-values compare AUC against the fusion model.

| Model | Accuracy | Sensitivity | Specificity | F1 | AUC (95% CI) | $p$ |
|---|---|---|---|---|---|---|
| Fusion | 0.772 | 0.725 | 0.820 | 0.761 | 0.861 (0.826–0.895) | Ref. |
| Macula | 0.752 | 0.795 | 0.710 | 0.763 | 0.832 (0.792–0.870) | 0.038 |
| Vessel | 0.708 | 0.685 | 0.730 | 0.701 | 0.772 (0.724–0.816) | <0.001 |
| Disc | 0.650 | 0.465 | 0.835 | 0.571 | 0.754 (0.705–0.799) | <0.001 |

**Regional attention weights.** The fusion module assigned mean weights of 0.408 (macula), 0.383 (disc), and 0.209 (vasculature); the macula–disc difference was not significant ($p = 0.506$), while vasculature was significantly lower ($p < 0.001$), confirming that macular and disc compartments jointly dominate the fusion decision (Korot et al., 2021; Betzler et al., 2021; Ilanchezian et al., 2021).

**Weight distributions by sex and outcome.** In correctly classified female samples, macula weights were higher relative to misclassified cases; in correctly classified male samples, optic disc weights were elevated (Figure 2).

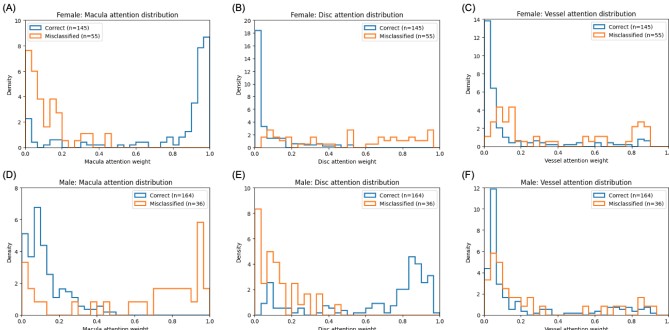

Figure 2: Distribution of branch-level attention weights stratified by biological sex and classification outcome

## 4. Discussion and Conclusions

We propose a multi-branch attention framework that enables quantitative, reproducible estimation of region-wise model reliance through ROI-level scalar weights with associated uncertainty estimates. Macular and optic disc compartments carried dominant discriminative signals, while vasculature was supplementary. Future work should examine whether this framework is applicable to region-wise attribution of other retinal biomarkers.

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
