# OpenReview forum: "Quantifying Regional Contributions to Sex Classification from Fundus Photographs via a Two-Stage Attention-Based Deep Learning Approach"
_MIDL.io/2026/Short_Papers — MIDL 2026 - Short Papers Poster_

### Official Review · Reviewer_MKyz · 2026-04-25
**Interesting study in general, but I am not sure if this paper really fits MIDL.**

**Rating:** 3
**Confidence:** 5

**Review:**

Interesting study in general, but I am not sure if this paper really fits MIDL. It does not have any relevant technical novelty and the analysis conducted is rather simple and the discussion of the results lacks depth. I would have for example expected to see a more thorough discussion of the accuracies reached by the single-ROI classifiers in relation to the attention weights. It remains also unclear why I would need such an attention-based approach to do what the authors are after. I think a similar result could have been achieved by just using saliency scores from a standard CNN. Again, this does not mean the paper is bad. I think it might be a better fit at an ophthalmology conference.

**Summary:**

The paper tries to shed light on which parts of the retina contribute most to the classification of sex from fundus photographs. This is done by selecting patches/ROIs around specific landmarks and feeding them into a attention-based classifier. Results show that macula and optic disc are the most important anatomical cues.

**Strengths:**

- Interesting and relevant problem
- Sound technical setup
- Evaluation using sex-balanced data
- Easy to read and follow
(no additional strengths)
(no additional strengths)
(no additional strengths)
(no additional strengths)

**Weaknesses:**

- No real technical novelty, standard approach
- Rather simple technical setup
- Discussion of results lacks depth
- Architectural choices remain unclear (CNN could achieve the same with ROI-based saliency scores)

**Justification Of Rating:**

Not a bad paper, but I am not sure if it is a good fit at a rather technical conference.

---

### Decision · Program_Chairs · 2026-05-08

Accept (Poster)